# Heavy Metal Health Risk Assessment in *Picea abies* L. Forests Along an Altitudinal Gradient in Southern Romania

**DOI:** 10.3390/plants14060968

**Published:** 2025-03-19

**Authors:** Constantin Nechita, Andreea Maria Iordache, Carmen Roba, Claudia Sandru, Ramona Zgavarogea, J. Julio Camarero

**Affiliations:** 1Department of Biometry, National Institute for Research and Development in Forestry “Marin Drăcea”, Calea Bucovinei, 73 bis, 725100 Câmpulung Moldovenesc, Romania; 2National Research and Development Institute for Cryogenics and Isotopic Technologies—ICSI Rm. Valcea, 4 Uzinei Street, 240050 Valcea, Romania; claudia.sandru@icsi.ro (C.S.); ramona.zgavarogea@icsi.ro (R.Z.); 3Faculty of Environmental Science and Engineering, Babes-Bolyai University, 30 Fântânele Street, 400294 Cluj-Napoca, Romania; carmen.roba@ubbcluj.ro; 4Instituto Pirenaico de Ecología (IPE-CSIC), Avda. Montañana 1005, 50080 Zaragoza, Spain; jjcamarero@ipe.csic.es

**Keywords:** heavy metals, forest ecosystems, mole ratio, atmospheric pollution, ecosystem health risk assessment, Romania

## Abstract

Heavy metals (HMs) from industrial pollution are bioaccumulated in plant tissues, but we lack information on their spatial variability in forest ecosystems. *Picea abies* L. needles, bark, and litter were collected at 17 sites along a 1000-m-wide altitudinal gradient in southern Romania to measure concentrations of mineral nutrients, non-toxic metals, and toxic metals. Isotopic ratios (^206/207^Pb, ^87/86^Sr) were used to infer contamination origins. We found significant differences in needle versus bark and litter elements’ concentrations, indicating that needles are sensitive biomarkers in tracking air pollution. We found high Fe, Mn, Zn, and Cu concentrations, which can be involved in the low Na, Mg, and P content in needles. The mole ratios indicate a negative relationship with As concentrations in needles. Several environmental health and ecological risk assessment indices document that Cd levels can represent a moderate risk for most sites. Our study shows that *P. abies* presented an elevated bioaccumulation potential for Fe, Zn, Cu, Ni, and Cr, respectively, and it only absorbed Na, Sr, Cd, and Li. The methodology and results presented herein may serve as a reference for future studies and provide a foundation to develop management strategies to mitigate heavy metal pollution in forest ecosystems.

## 1. Introduction

Atmospheric pollution is a significant environmental concern with far-reaching consequences in highly industrialized areas, urban settlements, and terrestrial ecosystems [1,2,3]. The UN Environment Programme (UNEP) estimated that unhealthy environments account for 12.6 million deaths yearly, with other findings indicating that 8.34 million deaths are solely due to fine particulate and ozone air pollution [4]. Most environmental research has focused on assessing pollution sources and various assimilation patterns in urban or industrial regions, evaluating multiple matrices such as soil, water, and vegetation and their transfer to the food chain [3,5,6,7]. In forest ecosystems, rising air pollution is less studied, even though it is not negligible; it has been documented that heavy metals (HMs) negatively affect forest productivity and tree growth, leading to dieback [5,8,9]. It is commonly assumed that forests are less affected than urban locations and that toxic levels remain below the detection limit, which is not always accurate, as significant contaminant enrichments have been discovered in numerous ecosystems worldwide [10,11,12]. Even so, the interaction between HM and the plant community under rising temperatures and drought is attracting attention due to the complex, interrelated processes involved [1,2].

Climate change also influences large-scale atmospheric deposition of contaminants through very high-pressure air masses responsible for long-way transboundary HM transport and deposition [3,4,5,6]. Intercontinental air masses can be vectors for transporting large amounts of hazardous dust particles charged with metallic ions from highly industrialized countries to remote areas [7,8,9,10]. However, the dissemination paths of contamination are not always well understood, particularly in forested areas where pollution levels are restricted to local sites. Previous reports have identified the origins of detected contaminants, which include deposits from industrial extraction and tailing dumps, activities related to ore processing, emissions from engines, combustion of fossil fuels, disposal of waste, and facilities used for waste incineration that contribute to atmospheric contamination [11,12,13,14]. The most common tool for tracing origins of metal pollution and its transfer function from the environment to the plant is ^206/207^Pb and ^87/86^Sr isotopic ratios [15,16,17]. Environmental pollution is mainly focused on toxic metals and metalloids such as cadmium (Ca), arsenic (As), chromium (Cr), and nickel (Ni), having densities below 5 g/cm^3^ and various properties such as persistence, potential for non-degradation, and waste, leading to risks for both humans and the environment [18].

Along with toxic elements, atmospheric pollution carries macro- and micronutrients such as K, Mg, Ca, and Na, which are essential in small amounts for the structure and functioning of prokaryotic cells but can be harmful in broad concentrations [19,20]. Despite this, the definition of plant nutrients has evolved, primarily encompassing two key components: structural and physiological significance [21,22]. Some authors separate essential nutrients from functional elements [23]. Mineral elements, such as sodium (Na), are often overlooked as essential for plant growth and development, even though Marschner’s classification, also adopted by European Union regulation, recognizes their importance [24]. Assessing mineral concentration in soil and plants involves evaluating mole ratios between different elements as a metric for estimating the nutritional quality of plant species [25]. Mineral ratios in dry matter are indicators of homeostasis in plants, which can illustrate hidden deficiency and an excess of one or a suite of elements [25,26,27]. A negative correlation of K/P, Ca/P, and Mg/P ratios with Cd content in soil and plant tissues was used to document the activation of the antioxidant system capable of reducing proline accumulation and inducing deficiency in the nutrient mole ratio [28]. Nutrient ratios can vary substantially under different transfer conditions from soil to plant biomass depending on growing season environmental conditions and phenological plant phases [29].

In the study area, forests are dominated by Norway spruce (*Picea abies* L.), a major European conifer. We chose this location to assess (i) whether there is significant contamination threatening environmental health and (ii) whether there are possible imbalanced nutrient mole ratios reflecting forest dieback in the region. To fulfill these aims, we evaluated essential nutrients (Ca, Mg, Na, K, P), d-block transition metals (Mn, Fe, Zn, Cu), and toxic metals (Cd, As, Li, Cr, Ni, Sr) in litter and Norway spruce needles and bark. Two isotopic ratios (^206/207^Pb, ^87/86^Sr) were used to investigate the possible origins of metals in tree organs and litter. The mole ratios Ca/P, Mg/P, Ca/Mg, K/Mg, Ca/Na, Ca/P, Mg/P, Ca/K, Na/Mg, K/Na, Li/Ca, and Li/Na were calculated and used as proxies for trees’ physiological alterations. Finally, we performed an ecosystem health risk assessment to document possible issues on trees and deliver information on ecological integrity and ecosystem service demand in the study area.

## 2. Results

### 2.1. Elemental Chemical Composition in Needles, Bark and Litter

There were no significant differences in the average elemental chemical concentration between control sites and the evaluated altitudinal transect, highlighting the widespread pollution caused by the atmospheric deposition range. Therefore, all sites were analyzed collectively, and the findings were compared to other reference studies. Between matrices, the concentrations decreased in this order: needles > bark > litter (Figure 1). However, Sr, Cd, and Li concentrations were higher in litter than in needles and bark, and Na concentration decreased from bark to litter and needles. An exception was As, which had above-detection-limit content only in needles. The elemental concentration differed statistically between needles, bark, and litter (Bonferroni test, test sign) only in the case of Ca, Mg, K, and Sr (Table 1). Except for Na, Sr, and Cd, all elements’ concentrations significantly varied between litter and needles. High variance (Levene’s test) was found for most elements, such as Ca, Mg, K, P, Fe, Zn, Cu, Ni, As, Cr, and Li. This result indicated that metals are randomly deposited in different quantities depending on environmental conditions, metal weight, transport distance, weather, and forest vegetation from the pollution source. The RMSE showed lower performance for the regressor model in almost all cases, which suggests that the variability is too high to be predicted with accuracy.

The altitudinal trends of each chemical element showed different patterns between matrices (Appendix A). For instance, Ca, Mg, K, and P had an up-and-down trend until 900 m, and Na progressively increased upwards. Around 1000–1100 m, in needles, the concentrations were high and relatively stable, which was opposite to bark. The mineral composition of litter decreased until 1200 m, followed by an increase in concentration, except for Mg and Na concentrations, which peaked at 1036 m (Appendix A). Non-toxic metals illustrate comparable trends between needles, bark, and litter, showing a high variability before 800 m, followed by a reduced concentration and slowly decreasing to the top of the mountain (Appendix A). An exception was Fe and Cu concentrations in needles, where values were higher, and the trends showed similar peaks at 1500 m, whereas Zn increased after 1300 m. The different altitudinal trends can indicate origins other than those from litter. The toxic elements disclosed more variability of trends with less synchronicity between matrices (Appendix A). Alkali metals such as Li and Cr peaked around 1100 m in needles, which was not found in litter and bark; however, their trend decreased with altitude in litter. We observed a peak in bark Li concentrations at 1300–1500 m. A clear decreasing trend of As concentrations was found as altitude increased.

### 2.2. Isotopic Signature and Mole Ratios of Ca, K, Na, Mg, P, and Li

The ^206/207^Pb values had a range interval of 1.16–1.20 (needles), 1.15–1.17 (bark), and 1.24–1.16 (litter), respectively, with mean values of 1.18 ± 0.01; 1.16 ± 0.00; 1.14 ± 0.01 (Figure 2). The altitudinal trend reflects an up-dawn trend until 900 m and a relatively stable trend until 1300 m, followed by a peak at 1500 m. The ^87/86^Sr values varied between 0.71–0.83 (needles), 0.71–0.78 (bark), and 0.71–0.73 (litter), and the averages were 0.75 ± 0.34; 0.73 ± 0.02; 0.71 ± 0.00. As the altitude increased, the trend of ^87/86^Sr was comparable to that of ^206/207^Pb, but only in the case of Norway spruce needles. The bark value was extremely low until 1200 m and peaked at 1330 and 1500 m in the litter, whereas Sr showed less variability between 1000 and 1300 m.

Except for Ca/P and Mg/P ratios, which did not significantly differ between litter and needles, the ratios showed broad distributions (Figure 2 and Table 2). The Ca/Mg and K/Mg ratios had a quasilinear trend between 850 and 1300 m in litter, and extremely low values were found at 650 and 1036 m. The Ca concentration contributes to higher Ca/Na ratio values in needles than in litter. Two Ca/Na peaks were observed in needles at 1160 and 1554 m. This ratio reflected different patterns since peaks were found at 654, 850, and 1180 m in needles and only at 1160 m in litter. There was a small Ca/K ratio in litter and needles. Still, the ratio documented a sustained accumulation of K in the litter, even though the needle metal concentrations were below the recommended limits. Peaks in the Ca/K ratio in litter were found at 850, 907, and 1110 m with a decreasing altitudinal trend. Similar results were found for the Na/Mg and K/Na ratios in the litter, which were relatively low and stable with altitude. In contrast, the Mg and K in needles induced a higher mean ratio with an increasing trend and peaks at 1110, 1160, 1310, and 1554 m. Li/Ca and Li/Na ratios were higher in litter than in needles.

### 2.3. Correlation and Multivariate Analysis

The multivariate analysis illustrates that the three most representative elements changed from litter (Ca, Li, Na) to bark (K, Li, Cu), and to needles (K, Na, Cu) (Figure 3a–c). Even so, the metals related to litter samples were not found in similar associations with bark or needles (Figure 3d–f). Exceptions were K vs. Mg and Ca, which had strong positive relationships across all matrices.

The mole ratios in litter showed more significant relationships than those in needles (Figure 4a,b). The litter ratios had a negative relationship between Li_bark_ vs. Ca/P, Cd_needles_ vs. Ca/K, Cr_litter_, and As_needles_ vs. Na/Mg. An opposite, strong positive relationship was found between Ca/Na, Ca/P, Mg/P, K/Na, and Li/Na, and Ni, As, Cr, and Li across all matrices (Figure 4a). The Ca/Mg and Ca/K ratios in needles decreased with rising As concentration (Figure 4b). A symmetrical relationship was found for Cr in needles and bark, demonstrating similarity to Li/Ca and Li/Na ratios.

### 2.4. Environmental Risk Assessment

A bioconcentration factor index (BCF) value > 1 indicated that needles and bark showed potential for metal accumulation. In our case, in needles, BCF had extreme values for Ca, Mg K, P, Mn, Fe, Zn, Cu, Ni, and Cr of 62, 36, 149, 126, 23, 511, 289, 59, 1399, and 2423 mg kg^−1^, which were found in 82, 76, 100, 58, 35, 100, 58, 64, 88, and 100% of samples. In bark, Ca, Mg, K, P, Mn, Fe, Zn, Ni, and Cr had maximum values of 23, 22, 88, 32, 10, 10, 111, 146, and 82 mg kg^−1^ found in 29, 29, 88, 23, 5.8, 5.8, 29, 64, and 47% of samples. BCF ≤ 1 indicated that plants only absorb and do not accumulate the metal in their organs, as was the case for Na, Sr, Cd, and Li (64, 82, 58, and 100% of samples) in needles, and Na, Sr, Cu, Cd, and Li (41, 88, 5.8, 100, and 100% of samples) in bark (Figure 5).

The sites were classified as unpolluted or considerably contaminated when evaluating the litter. The Geo-accumulation index had negative values except for Cd and Li, where the variation interval was between −0.09 and 1.36 and −1.31 and 1.18 mg kg^−1^ (Figure 6a). Contamination factor values indicated moderate contamination levels, particularly for sites located at 654, 907, 950, 1110, and 1310 m altitude (Figure 6b). Ecological risk was below 1 for Mn, Zn, Cu, Ni, and Cr, but in the case of Cd, the mean values were 74 mg kg^−1^, i.e., moderate values above 100 were found at 907, 950, and 1310 m, corresponding to considerable enrichment (Figure 6c). The pollution load index showed uncontaminated litter ranging between 0.01 and 0.11 mg kg^−1^ in all cases. The Nemerow pollution index had values between 2.74 and 2.78 mg kg^−1^, indicating moderate contamination. Finally, ecological risk and potential ecological risk values documented a low risk associated with toxic elements in litter (Figure 6d).

## 3. Discussion

### 3.1. Elemental Distribution, Origins and Altitudinal Trends

The bioaccumulation of HMs and other atmospheric pollutants in tree organs is highly important for evaluating ecosystem health [30,31]. The origins of heavy metals in the Cozia Mountains are mainly anthropogenic, resulting from two industrial facilities and road traffic. Similar values of ^206/207^Pb and ^87/86^Sr isotopic fingerprints were obtained in water and sediment samples in the Olt River that crosses our study area, which was demonstrated to be from anthropogenic inputs [32]. All metals showed a high variability unrelated to altitude, except for As in needles, which showed a decreasing trend (Appendix A). We found that nutrient content had sustained high values between 950 and 1160 m, which decreased until 1310 m in needles; the opposite pattern was found in bark, which peaked around 1023 m in litter. Also, the non-toxic and toxic elements presented intermetallic similarities rather than within matrices, which can be attributed to particle weight [33]. Air currents from higher altitudes can transport significant volumes of particles charged with metals [34]. Earlier studies on long-range HMs atmospheric transport illustrated that remote high-elevation forest limits and treelines have been affected at least since the 1980s [35,36]. The process of transporting atmospheric deposition is complex and depends on anthropic activities that generate emissions, particle size, climatology, topology, and forest characteristics. High-elevation forest limits are exposed to specific wind and abrasion contact forces with air masses, causing differences in metal deposition [37]. Both dry and wet deposition can be factors of different metal deposition and absorption since water-soluble metals (Ni, Fe, Pb, Mn, Cd, Zn, and Cu) are reflected in vegetation and soil solution [38]. In comparison, insoluble metals (Cr, Ni, Fe, Al) are mainly found in vegetation layers [39,40,41]. Point sources and the distance to the affected area are essential to systematic contamination [42].

The presented results indicated high bioavailability in Norway spruce, mainly for Zn, which decreases with distance from a point source. These findings were also presented in other similar studies [43]. It was noted that in Norway spruce bark, the chemical concentration content differs between 0.25–0.45 mm and below 0.18 mm granulometric fraction, being higher in the fines fraction [44]. The Cu (5.5 mg/kg), Zn (155 mg/kg), Ni (4.28 mg/kg), and Cr (2.11 mg/kg) obtained in a natural forest from Sweden can be compared with our mean bark concentrations [44]. A similar study in Sweden indicated higher concentration (mg/kg to evaluate HM content in the litter) in bark of Ca (7290), K (2490), Mg (1110), P (640), and Na (96), compared with our results, but similar values of toxic elements Sr (31.1), Cr (0.47), Ni (2.94), and Cd (0.63) [45]. Our results show that the heavy metal analysis of the bark had lower concentrations than in the needles, which can be used to conclude that needles can be used as bioindicators of heavy metal pollution and the status of elements essential for vital growth. Similar results have previously been presented in the literature [46,47].

The metal concentration in needles is used to assess nutrient deficiency under pollution stress, offering valuable information regarding the contamination degree and the mineral levels for sustaining vital plant growth [48,49]. The mean concentration of essential nutrients in needles had a decreasing order from K > Ca > Mg > P > Na, with maximum values of 7200, 6300, 940, 971, and 107 mg/kg (Figure 1; Table 1). The limits for plant nutrition and global health include values (mg/kg dry weight) between 500 and 40,000 (K), 500 and 10,000 (Ca), 2000 and 5000 (Na), 1500 and 3500 (Mg), and 2000 and 5000 (P) [50]. According to the present guideline, only Ca and K are within the limits, and Na, Mg, and P are much below them. The minimal nutrient requirements for Norway spruce normal growth and development are mentioned in the literature as >6000, 700, 800, and 1800 mg/kg for K, Ca, Mg, and P [51]. In this case, our findings suggest that Mg is within the minimal requirements, and only P is below the beneficial threshold. Phosphorus is mainly responsible for maintaining the normal water regime, and its deficiency is correlated with high N deposition [52]. Elevated N availability reduces carbon allocation to roots and ectomycorrhizal fungi, depleting its retention and increasing leaching, causing tree functional and growth issues [53]. Additionally, a recent monitoring study of the mineral content of Norway spruce needles shows that P deficiency has been chronically declined in the last 25 years on Ore Mountain, possibly associated with environmental pollution [54]. Mg is essential in chlorophyll biosynthesis and is mobile, being reported to be translocated in Norway spruce from older to younger needles [55]. Mg seems to induce sensitivity to photosynthesis in forest ecosystems and regulates biomass production under scarce resources [56]. The Mg from air pollution enrichment has low transfer ability from the bark and needles to the wood [57].

The second group of chemical elements evaluated was d-block transition metals, which, according to our results, had a decreasing order in all matrices as follows: Fe > Mn > Zn > Cu (Figure 1; Table 1). Comparison of measurements performed on our samples with optimal concentration ranges as necessary for Fe (50–150 mg/kg), Mn (10–20 mg/kg), Zn (15–30 mg/kg), and Cu (1–5 mg/kg) [51] revealed above-sufficiency threshold values. The measured Fe concentration ranges between 1000 and 6599 mg/kg, which exceeds the thresholds by over 43 times and is associated with a competition mechanism with other essential elements. Higher metal content, such as Fe, Mn, Zn, and Cu, detected in needle samples can be attributed to their role as significant components of emissions from industrial activities and vehicle engines. The needles of pine collected from urban areas exposed to street dust pollution had Fe concentrations up to 29,465 mg/kg [58]. In similar studies regarding urban pollution and natural forest ecosystems, Fe had values in needles up to 274 mg/kg [59], 248 mg/kg [60], 356 mg/kg [61], 189 mg/kg [62], and 644 mg/kg [63]. High Fe ions can induce oxidative stress, damaging the cells and plasma membrane and increasing the production of reactive oxygen species. Mn is a nutrient with physiological implications since it acts as an activator and cofactor of various metalloenzymes [64]. Still, under normal conditions, concentrations above 400 mg/kg can be toxic for most plants [65]. When evaluating similar studies, we found a wide range of concentrations: 1804 mg/kg [66], 720 mg/kg [67], 591 mg/kg [68], and 117 mg/kg [69], concluding that the maximum Mn concentration of 1033 mg/kg found in our study can be detrimental to Norway spruce. Zn is mainly assessed as a nutrient that increases resistance against drought stress by regulating physiological and molecular functions such as cell membrane stability, stomatal regulation, and water use efficiency, and as a stimulator of antioxidant activity [70,71]. The concentration of Zn in tree needles is reported to be 56 mg/kg [68], 570 mg/kg [72], and 47 mg/kg [73]. In an urban area, *Picea punges* were used as bioindicators of urban air pollution, and the results indicated that the Zn concentration depends on traffic density and the needles’ ages, with values up to 80 mg/kg in 5-year-old samples [74]. Cu is essential for enzymatic activities, chlorophyll production, respiration and antioxidant systems, and signal transduction [75]. The Cu concentration decreases with needle age, and translocation to younger parts of foliage is observed [76]. The previous reports show values of Cu in needles of 26 mg/kg [77], 19 mg/kg [78], 9.2 mg/kg [68], 7.7 mg/kg [76], which show that in our study area, the values are not excessively high but can be associated with road pollution.

The third group of toxic metals shows a decreasing order of the mean concentrations in needles from Sr > Cr > Ni > Li > As > Cd (Figure 1; Table 1). The interval range for normal growth and development in plants was set at Cr (1–2 mg/kg), Ni (0.1–1 mg/kg) [50], Cd (0.098 mg/kg) [79,80], As (5 mg/kg), Sr (350 mg/kg), and Li (20 mg/kg) [81]. Strontium (Sr) may occur naturally; however, in this case, it can be associated with industrial activities conducted in Copsa Mica, leading to severe groundwater contamination [2]. We found a maximum 81 mg/kg concentration at the 1160 m altitude site. The results are comparable with other European reports, which indicated values of 63 mg/kg [82], 71.9 mg/kg [83], 33 mg/kg [69], and 37 mg/kg [84]. Sr is an alkaline earth metal with a 15-ranking abundance (340 ppm). The Sr exhibits mobility and can translocate between various plant organs via the plasma membrane transporter mechanism for Ca and K. Those elements are replaced, and stable Sr induces several toxic effects, including reduced growth rates [85]. Chromium (Cr) varied between 19 and 127 mg/kg, with the maximum value at the 1160 m altitude site. The recommended values in plants were exceeded at all sites, and the content was much higher compared to similar results in needles exposed to traffic, as is the case of 4.5 mg/kg [69], industrial urban areas 16 mg/kg [78], or a natural protected park 10.6 [86]. Cr is the 21st most abundant element in the crust (100 ppm) found in air particulate matter and is registered as a priority pollutant uptake by plants through phosphate or sulfate transporters [87]. Several toxic effects of Cr on plant physiology, biochemistry, molecular traits disrupting the cell cycle, enzyme activity, nitrogen assimilation, and antioxidant activity, were previously reported [40,88,89]. Nickel (Ni), in small amounts, is beneficial for regulating metabolism through enzymatic activity, with values ranging from 4.3 to 36 mg/kg, significantly higher than the recommended limits. The maximum levels were measured at altitudes between 521 and 723 m. Other results show lower concentrations of 1.1 mg/kg [69], 6.4 mg/kg [68], 1.3–5.2 mg/kg [78], and 9.78 mg/kg [86]. Even though some authors suggested that values up to 10 mg/kg are toxic for sensitive species [50], in low-polluted areas, reported values were between 8.1 and 55 mg/kg, mentioning that Ni bioavailability increased with soil acidity and organic matter content [66]. Lithium (Li) is a metal of great interest in transitioning to green energy, increasing yearly demand and consumption, causing extensive waste worldwide, and inducing toxic effects on plants by affecting photosynthesis, DNA biosynthesis, and enzyme activation [90]. In higher amounts, it can displace K^+^, Mg^2+^, and Ca^2+^ in biological membranes [91]. In our study, Li varied between 5.2 and 37 mg/kg (maximum values around 1000 m and 521 m altitude), comparable with the average of 30 mg/kg reported for the earth’s crust [90,92]. In plants, Li concentrations have not been extensively studied, and the thresholds for severe toxicity up to 40 mg/kg in most cases differ between species, whether naturally occurring or resulting from contamination [93]. Similar results show values of Li in needles ranging from 14 to 48 [94], 0.03 to 0.07 mg/kg [25], and 4.0 mg/kg [69]. Arsenic (As) is a highly recognized metal with toxic potential originating from natural sources and anthropogenic activities, and it is easily transported through air and water [95]. Here, we found values ranging between 0.37 and 0.87 mg/kg, comparable with other findings of 0.20–0.31 mg/kg [25], 0.53 mg/kg [69], and extremely high values were also reported in tailing dumps of an abandoned lead-zinc mine in South Korea (11.7 mg/kg) [95]. The highest concentrations are found between 500 and 1000 m altitude. Cadmium (Cd) is extensively studied for its harmful effects on living organisms, with one of the primary sources of contamination being atmospheric deposition [96]. The higher toxicity increases reactive oxygen species, causing membrane damage and cell biomolecule degradation. Cd reduces Fe and Zn uptake in plants [97]. Still, high chlorophyll, carbon, hydrogen, nitrogen, phosphorus, potassium, proline, soluble sugars, and phenolic compounds in *Pinus sylvestris* and *Abies alba* needles increase photosynthetic functions and enhance Cd tolerance [8,67]. In our study, the range of Cd varied between 0.11 and 0.41 mg/kg, which is not extreme. Similar values were previously reported as 0.11 mg/kg [86], 0.23 mg/kg [78], 0.18 mg/kg [80], 0.036 mg/kg [69], and 0.08 mg/kg [25].

### 3.2. Relationship Between Nutrients Mole Ratio and Toxic Metals

Atmospheric particulate matter (dry or wet deposition) through soil–plant systems is included in the terrestrial biogeochemical cycle and can induce climatic effects by absorbing or scattering solar radiation and cloud condensation nuclei [98]. Unfortunately, few studies on tree species in natural forests evaluate the toxicity of the deficiency of various mole ratios compared with stoichiometry in Norway spruce, with more results reported for *Larix* sp. and *Pinus* sp. [25,99,100]. Even so, the literature indicates an optimal mole ratio of Ca/Mg (6.5:1), Ca/K (13:1), Mg/K (2:1), Ca/P (1–2:1), and K/Mg (1.5:1) [101,102,103]. The Ca/K (0.85) and Mg/P (2.05) ratios in needles found in our study reflect a balanced mineral ratio, also discussed in similar studies [99]. Additionally, in needles, we calculated mole ratios of K/Na (81:1) and Ca/Na (70:1), revealing an increase in the proportion of essential elements potassium and calcium. In our study, K and Ca are not in elevated amounts, and their deficiency is associated with detrimental effects on photosynthesis, osmoregulation, and several other essential processes. Sodium, in high concentrations, is harmful for physiological processes by interfering with K^+^ uptake, which is not the case in our study, where it was below the threshold and considered beneficial. The dualism of K/Na and Ca/Na can affect biophysical and biochemical processes, metabolic cell functions, activation enzymes, respiration, and photosynthesis, but only when Na is in higher amounts [104]. K and Mg are involved in photosynthesis, phloem transport of photoassimilates, and photoprotection [105], and in our study, the ratio was 7:1, which can be considered normal. The literature indicates that the high concentration of certain metals, such as Fe and Mn, in needles could explain the deficiency in essential minerals [48,49]. Several toxic metals, such as Cr, Ni, Cd, and As, are associated with Na, Mg, and P deficiency in needles [106]. Based on correlation analysis, an increase in toxic Zn reduces K and Ca in needles. Additionally, the increasing As in needles negatively affects Ca/Mg and Ca/K, a fact sustained by correlation and hierarchical cluster analysis (Figure 3c and Figure 4b). Arsenic inhibits the uptake of various nutrients, including Ca, Mg, and K, since it can enter the cells of plants using channels of essential nutrients [107,108]. A more complex relationship between mole ratios and toxic metals is observed in litter, suggesting that toxic metal concentrations increase during litter decomposition. The literature reveals gaps in information regarding changes in the elemental composition of litter after decomposition. However, the limited results also showed that the concentrations of Cr and Cd increased, regardless of ecosystem type and litter species during decomposition [109].

Recent advances have pointed out the importance of studying Li isotopic fractioning during watering and its content vs. mineral ratios to assess past environmental processes that involve weathering and hydrothermal circulation [110,111,112]. Li mobility is related to critical cations in the carbon cycle (Ca, Mg, Na, K, P), which vary over time but can evolve quickly and have similar mobility in all elements [113]. These results suggest that the Li and ^7/6^Li isotope ratios can precisely trace Earth’s surface processes [114]. A recent country-level database showed that Li/Ca and Li/Na in natural waters (0.035–3.02 and 0.005–0.21) [111] differed from those measured in needles (0.002–0.008; 0.05–0.66) and litter (0.02–0.75; 0.07–0.82). The Li/Ca and Li/Na ratios are temperature proxies, and they differ significantly between abiotic and biogenic components due to elemental assimilation during carbonate mineral precipitation under soil acidity and dissolved inorganic carbon [115]. Significant peaks were found for both ratios at 650, 950, and 1100 m, which can be explained by a low decomposition rate in litter and transfer to soil rather than plant assimilation, reflected by extremely low BCF indices.

### 3.3. Environmental Health Risks

The bioconcentration factor (BCF) indicates the potential to uptake single HMs and translocate them into tree organs. Our results documented an imbalanced contamination level between needles and litter, with high BCF values in needles and low transfer from litter to plant organs. The average BCF of Fe, Zn, Cu, Ni, Cd, Cr, and Li was 164, 36, 14, 360, 1, 484, and 0.4 in needles and 3.2, 17.6, 2.8, 53, 0.4, 23, and 0.1 for bark, respectively. Needles are more suitable for monitoring metal pollution since they have a higher accumulation ability than bark. The results indicate that Norway spruce had a higher accumulation capacity of Cr, Ni, Fe, Zn, and Cu. The litter was classified as unpolluted or considerably contaminated, but the imbalanced metal levels can be a problem for forest growth. The Cd concentrations induced a moderate ecological risk at most sites. Even so, at 907, 950, and 1310 m altitude, the ecological risk is considerable in our study area, necessitating future monitoring activities due to its high toxicity. The I_geo_, C_f_, PLI, PI_Nemerow_, and PERI indicate that Cd can represent a moderate risk for most sites.

The integrated assessment of pollution in biological samples documents two aspects: (i) mixed sources of contaminants transported from sap flow to leaves originating from inorganic and organic soil used to assess past pollution, and (ii) stomatal absorption, which discloses air deposition [116,117,118,119]. Together with minerals, toxic metals (Hg, Cu, Pb, Ni, Zn, As) are assimilated, affecting plant physiology [120]. Depending on its amount, each HM is responsible for detrimental effects on plant growth and development, and it can affect the functioning of the soil microbial community structure, altering C:N:P stoichiometry [121]. The As, Pb, Cr, Co, Mn, and Zn concentrations in litter induce inhibition of carbon fixation, resulting in negative implications for biogeochemical cycles [122,123]. Not only do absolute values of metal concentration in plants reflect dysfunctions between physiological processes but also balanced mineral composition.

## 4. Materials and Methods

### 4.1. Study Area and Sample Collection

The study area is located in the Southern Carpathians, in the “Cozia Mountains” National Park, which covers 17,100 ha (Figure 7a). The region is considered to be little affected by pollution and where the country’s most prominent resorts for treatment (pulmonary diseases) and relaxation are located. A highly used road crosses the park through the Olt Valley and connects the northwestern country border to Bucharest. The geological structure combines crystalline and sedimentary rocks, and several particular geomorphological characteristics are found along the altitudinal gradient. The highest mountain peak reaches 1668 m a.s.l. A succession of vegetation layers starts at the mountain base with European beech (*Fagus sylvatica* L.), mixed with oaks (*Quercus petraea* (Matt.) Liebl.) and birch (*Betula pendula* Roth). Upwards, *P. abies* and *Abies alba* Mill. become dominant. Two significant contamination sources were identified at less than 100 km from both edges of the Olt Valley, namely the Olchim chemical plant and the Copsa Mica Industrial plant (Figure 1). The Oltchim chemical plant was founded in 1966 to produce inorganic, macromolecular, and organic synthesis products (chlor-alkali, oxo-alcohols, propylene oxide, vinyl chloride monomer). This industrial complex includes the Sodic Govora plants that produce live steam, and contain facilities for producing, conditioning, and delivering fuels, water, electricity, and thermal energy. The principal emissions of metal pollution resulting from its activities include Cu, Zn, Pb, Mn, Cd, Ni, Co, Fe, and Hg [5,32,124]. The industrial activity in Copsa Mica began in 1935 and has caused significant environmental issues by releasing substantial amounts of Hg, Sr, Li, Pb, Cd, Cu, Al, Cr, Ni, Co, Fe, As, Zn, and Mn into the atmosphere, earning recognition as the most polluting source in Europe [2,125,126].

The sampling design was created to understand whether pollution differs along an altitudinal gradient. In mid-October 2021, we collected first- and second-year-old needles and tree bark from dominant, apparently healthy *P. abies* trees. In each sampling plot, samples were collected from three trees. Specifically, three plots were selected from the litter, with a distance of five meters between each plot. All materials were subsequently mixed, and a selection was further evaluated. Of the 17 locations, 3 are outside the national park, inside the mountain, with no evident contamination factor considered as control sites, and 14 are within the park, near a highly used national road (Figure 1). The 14 sites evaluated for pollution were near the Olt Valley, which also transports air masses possibly contaminated by two highly industrialized areas. We chose an intermediate southern exposition (inside the forest) on the roadside to encompass an almost 1000-m-wide altitudinal gradient. Samples were selected by hand from the first-level branches (2–3 m height) with rubber gloves, thoroughly mixed, and transported in plastic bags to the laboratory, where they were stored in a cool, dry space. In addition, we collected and investigated litter samples from the litter horizon, fermentation, and the humus horizon, depending on the altitude and degree of decomposition. In the litter horizon, the material was undissolved and in fermentation; and in the humus horizon, the material was decayed, exhibiting various degrees of decomposition. To assess heavy metal (HM) content within the litter, we collected samples from the uppermost layer of the forest floor, which contains mineral-organic debris, including needles, branches (less than 5 cm in diameter), fruits, and various types of plant matter exhibiting different stages of decomposition. The bark was extracted from the trunks of trees. In each area, we took six samples from trees of similar diameter and mixed the samples to reduce the environmental variability, georeferenced at the time of sampling. In total, 901 measurements were obtained from three matrices, considering 17 chemical elements.

### 4.2. Sample Preparation and Analysis

#### 4.2.1. Elemental Analysis

The collected samples were transported into plastic bags and air-dried at the laboratory at 80 °C for 24 h, and the dry weight was measured. The material was finely ground in an agate mortar before being sieved (<0.2 mm). A digestion process was applied to the intended samples to bring them into solution. A closed iPrep vessel speed iwaveJ system, MARS6 CEM One Touch (Microwave Accelerated Reaction System, CEM6 One Touch, CEM Corporation, Charlotte, NC, USA), was used for this procedure to decompose the organic compounds from soil and plant material and extract the target elements to assess multi-elemental metals. This was done according to the two-step temperature-controlled digestion program Microwave Digestion-CEM-Mars 6 Method Note Compendium, 2019. Samples were subjected to microwave-assisted nitric, chlorhydric, and fluorhydric acid digestion. Approximately 0.5–1.0 g aliquot of each sample was weighed, followed by digestion in a mixture of (3 mL HNO_3_ 69% + 2 mL HCl 38% + 5 mL HF 40%) at high pressure, temperature, and the combination of the microwave was increased up to 200 °C, during 6 min, and maintained at this level for 20 min. All plasticware was cleaned by soaking it for 24 h in 10% HNO_3_, using ultrapure water, then drying it and ventilating it in an oven at 40 °C. The vessels were cooled and carefully opened at the end of the digestion process. Each digest was transferred quantitatively with ultra-pure water to a 50 mL volumetric flask.

The Cd, As, Cr, Li, and Sr were measured in litter, needles, and bark using inductively coupled plasma (ICP-Q-MS Plasma Quant Elite, Analytik Jena, Jena, Germany) equipped with an AIM 3300 Autosampler (Analytik Jena, Jena, Germany) and collision reaction interface iCRI working in H_2_ and He modes. Calibration standard solutions were prepared by successive dilution of a high-purity ICP-multielement calibration standard (10 μg × L^−1^ from twenty-nine element ICP-MS standard, Matrix: 5% HNO_3_, ICP Multi-Element Standard Solution XXI CertiPUR^®^, Merck KGaA Frankfurter, Darmstadt, Germany). An internal standard was prepared from the dilution of 100 μg × mL^−1^ internal standard solution of Tb, Y, Sc, Bi in a matrix of 2% HNO_3_ (Anlytik Jena, Jena, Germany). Working parameters for plasma were chosen to obtain a good compromise between high sensitivity and low oxide levels (^140^Ce^16^O+/^140^Ce < 3.0% and ^137^Ba_2_+/^137^Ba^+^ < 3.0%). The following instrumental parameters of the ICP-Q-MS Plasma Quant Elite spectrometer were set: 1.00 L × min^−1^ nebulizer gas flow (NEB); 1.1 L × min^−1^ auxiliary gas flow (AGF); 9 L × min^−1^ plasma gas flow (PGF); 1200 W (ICP RF Power).

Mn, Ni, Cu, Zn, and Fe were quantified by flame atomic absorption spectroscopy (AAS). An AAS NOVAA 300 model, with Air-C_2_H_2_ flame type of an average fuel flow rate between 0.8–4.0 L × min^−1^ and the support gas flow rate between 13.5–17.5 L × min^−1^, was used for sample analysis. Hollow cathode lamps of the different metals were used as radiation sources, and the analytical measurements were based on time-averaged absorbance. Resonance lines at 279.5, 232.0, 324.7, 213.8, and 248.0 nm were employed for Mn, Ni, Cu, Zn, and Fe. Lamp intensity (4–6 mA) and bandpass (0.2–0.5 nm) were used. Air/acetylene flow rates were between 0.9–1.1 L × min^−1^ for all metals. Before each series of measurements, a calibration line was created for each element. Resonance lines at 422.7, 766.5, 285.2, and 589 nm were employed for macro elements such as Ca, K, Mg, and Na, using a lamp intensity (4–6 mA) and bandpass (0.2–0.5 nm). Air/acetylene flow rates were between 0.9–1.1 L × min^−1^ for all metals. All chemicals used were of analytical grade and were obtained from Sigma-Aldrich. All solutions and sample dilution for ICP-Q-MS analysis were prepared with deionized water obtained from a Milli-Q Millipore system (Bedford, MA, USA) with resistivity not less than 18.2 MΩ cm^−1^. The multielement standard solution CertiPur^®^ with a certified value of 1000 ± 3 mg × L^−1^ obtained from Merck, Germany, was used for the calibration curve in the quantitative analysis. For both analytical methods mentioned above, all the investigated calibration curves were characterized by a high correlation coefficient (*r* > 0.995), and a correlation coefficient of 0.9999 was a requirement when assessing the method performance of ICP-MS.

#### 4.2.2. Analyses of ^206/207^Pb and ^87^Sr/^86^Sr Isotope Ratios

The instrument performance of ICP-Q-MS Plasma Quant Elite was optimized for the elemental profile of Sr and Pb, and ^87^Sr/^86^Sr and ^206/207^Pb isotope ratios. A Sr isotope (25 µg L^−1^) standard reference solution (NIST SRM 987 Strontium Carbonate Isotopic Standard [127]) was used as the isotope standard. NIST SRM 987 and SrCO_3_ are certified for their Sr isotopic composition with an ^87^Sr/^86^Sr certified value of 0.71034 ± 0.00026 (2 σ) for the NIST SRM 987 Sr strontium carbonate isotopic standard reference material. The relative standard deviation value was 0.00046, with an RSD% of 0.0648%. A 0.1 g hot-dissolved Pb rod isotope standard reference solution (NIST SRM 981 common Lead Isotopic Standard [128]) was used as the isotope standard. The NIST SRM 981, Common Lead Isotopic Material, certified for its Pb isotopic composition with a ^207^Pb/^206^Pb certified value of 0.91464 ± 0.00033, demonstrated the value for an average of 0.91453, with a precision of 0.040418%. The accuracy and precision of the isotope ratio methods for Sr and Pb were tested using replicate ICP-MS Plasma Quant Elite measurements (Analytic Jena GmbH, Jena, Germany).

### 4.3. Statistical Analyses

Descriptive statistics were used to evaluate each dataset and to assess the subsequent statistical analysis. Pairwise comparisons between elemental levels in needles, bark, and litter were conducted using the Turkey post-hoc test, which was based on multiple correlations. Next, the normal distribution hypothesis and variance homogeneity in sample data were tested using Shapiro–Wilk and Bartlett’s tests, respectively. In addition, multiple linear regression, one-way variance analysis (ANOVA), and correlation analysis were calculated to evaluate the intrinsic relationship between macronutrient content and metals. The measure of regression fit was done by calculating the Root Mean Square Error (RMSE), interpreted as the standard deviation of unexplained variance; lower RMSE values indicate a better fit. The means comparison was performed using Bonferroni correction for significance level, and the Levene test was used to compare variances. A Spearman’s rank correlation analysis and hierarchical clustering analysis (HCA) were performed using group average and correlation distance type, showing the similarities on the y axes to assess possible relationships between elements and to understand their origins. Statistical computation was performed using Matplotlib v3.10.1, SciPy v1.15.2, Pandas v2.2.3, and NumPy v2.2.0 packages under the Python v3.13.1 environment [129,130].

### 4.4. Risk Assessment

Six risk assessment indices—one index for assessing heavy metal in plants (BCF), and five indices (I_geo_, C_f_, PLI, PI_Nemerow_, and PERI) for understanding the pollution degree in litter—were used to evaluate ecosystem health and to understand the possible effects of heavy metal contaminant air deposition. The intervals of interpretation are presented separately for each index in the Table 3.

The Bio-concentration factor (BCF) was assessed to determine whether trees can be classified as metal accumulators. The degree of chemicals absorbed from the environment in plant organs was evaluated as the metal concentration in a specific plant organ (bark, needles) at sampling time (C_p_) reported to litter concentration (C_s_) in mg/kg (Equation (1)).(1)$$BCF=C_{p}/C_{s}\;$$

The Geo-accumulation index (I_geo_) is a measure used to evaluate the amount of HM contamination in soil (litter) samples (in our case, in the litter) that considers the natural diagenesis process [131]. The I_geo_ index was calculated according to Equation (2).(2)$$I_{geo}={log}_{2}\;(C_{n}/1.5\times B_{n})\;$$
where C_n_ represents the concentration of metal in the soil (litter) (mg/kg), B_n_ represents the geochemical background, and 1.5 is a constant used to correct potential variation in the baseline data.

The Contamination factor (C_f_) is a measure to evaluate the contamination status of metals in the litter based on their measured concentration and a background value [132].(3)$$C_{f}=C_{m}sample/C_{m}background$$
where the C_m_ sample is the metal concentration in soil (litter), and the C_m_ background represents the metal amount from a natural reference considered naturally occurring in the continental crust.

The pollution load index (PLI) is a measure for evaluating the potential magnitude of heavy metal contamination in soil (litter) and was assessed using Equation (4) [133]. The PLI documents the contribution of each HM to regional pollution and can reflect temporal and spatial changes.(4)$$PLI={\left({CF}_{1}\times {CF}_{2}\times {CF}_{3}\times \ldots {CF}_{n}\right)}^{1/n}$$

The Nemerow pollution index (PI_Nemerow_) is a weighted multi-factor index that evaluates the quality of an environment by considering extreme values compared with the criteria imposed (Equation (5)).(5)$$P_{N}=\sqrt{({{(P}_{1})}^{2}+P_{imax}^{2})/2}\;$$(6)$$P_{1}=\frac{1}{n}\sum_{i=n}^{n}P_{i}$$
where P_N_ is the comprehensive pollution index of the sample, P_imax_ represents the maximum value of a single-value pollution index of the contaminants, and P_1_ is the average value of the single-factor index (Equation (6)).

The potential ecological risk (PERI) was used as a diagnostic tool for the HM concentration in the air deposition resulting from anthropogenic activities due to increasing elemental concentration in the litter and vegetation organs, which could threaten ecosystem health and functions. The integrated risk index is a concept that accumulates heavy metal characteristics, environmental behavior, and possible toxicological effects using an equivalent property index grading method [10,134]. The potential ecological risk index of a single element and for multiple elements was calculated using Equations (7)–(9).(7)$$C_{f}^{i}=C_{s}^{i}/C_{n}^{i}$$(8)$$E_{r}^{i}=T_{r}^{i}xC_{f}^{i}$$(9)$$RI=\sum_{i=1}^{n}E_{r}^{i}$$
where $C_{f}^{i}$ represents the pollution coefficient of a single element, $C_{s}^{i}$ is the value of litter heavy metal, $C_{n}^{i}$ is the background level, $E_{r}^{i}$ is ecological risk, and $T_{r}^{i}$ is the toxic response factor for a single element, which accounts for toxicity and sensitivity requirements.

## 5. Conclusions

We investigated the contamination level, altitudinal distribution, sources, and ecological hazards of heavy metals derived from air pollution and deposition in the “Cozia Mountains”, southern Romania. Overall, heavy metals originating from anthropogenic sources showed extremely high levels in needles compared to bark and litter. We used ^206/207^Pb and ^87/86^Sr isotopic fingerprints to document the origins of contaminants. Altitudinal trends were not observed, but evident changes in metal concentration around 1000 m, which varied between needles, bark, and litter, can be related to the transport and deposition by large atmospheric air masses. The results indicate that the Fe, Mn, Zn, and Cu levels exceed the recommended thresholds, which could negatively impact environmental health and interfere with essential elements such as K. Also, we observed that the Ca/Mg and Ca/K ratios were negatively correlated with As content in needles. Norway spruce can bioaccumulate high concentrations of Fe, Zn, Cu, Ni, and Cr in needles and bark, which is why it can be used for bioremediation strategies. The litter was classified as unpolluted to considerably contaminated, and at 907, 950, and 1310 m altitude, the ecological risk is considerable due to Cd pollution. Our results documented that contamination levels of long-range atmospheric deposition are detected and may affect the functioning of forest ecosystems.

## Figures and Tables

**Figure 1 plants-14-00968-f001:**
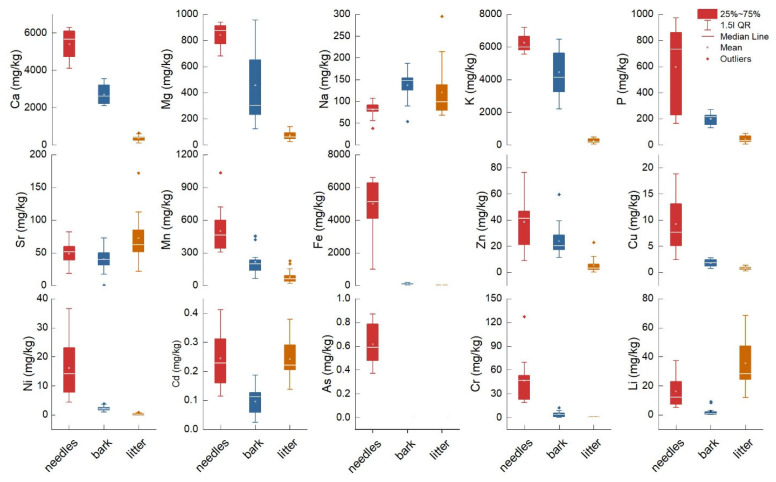
Elemental chemical concentration in Norway spruce needles, bark, and litter measured in Cozia.

**Figure 2 plants-14-00968-f002:**
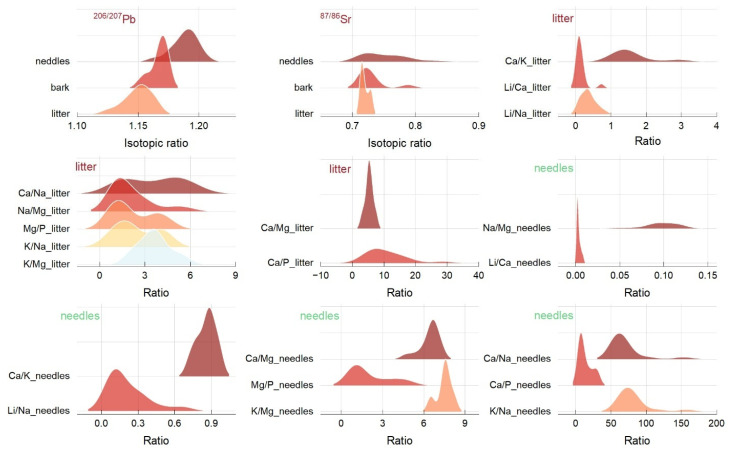
The distribution of isotope ratios ^206/207^Pb and ^87/86^Sr and ratios between nutrients measured in Norway spruce needles and litter.

**Figure 3 plants-14-00968-f003:**
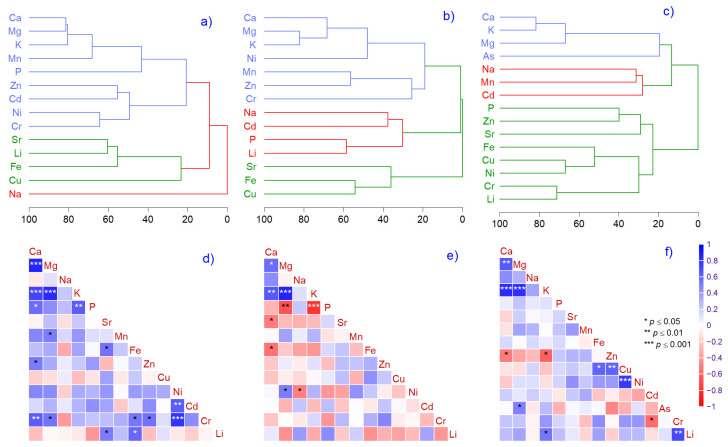
Hierarchical cluster analysis (**a**–**c**) and correlation analysis (**d**–**f**) of litter (**a**,**d**), bark (**b**,**e**), and needle (**c**,**f**) element concentrations. Asterisks indicate significance levels.

**Figure 4 plants-14-00968-f004:**
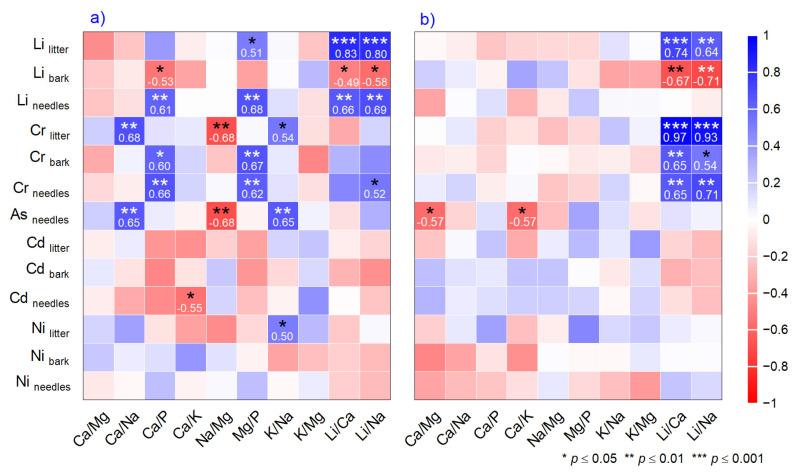
Correlations between mole ratios in (**a**) litter and (**b**) needles, and toxic elements in needles, bark, and litter. Asterisks indicate significance levels.

**Figure 5 plants-14-00968-f005:**
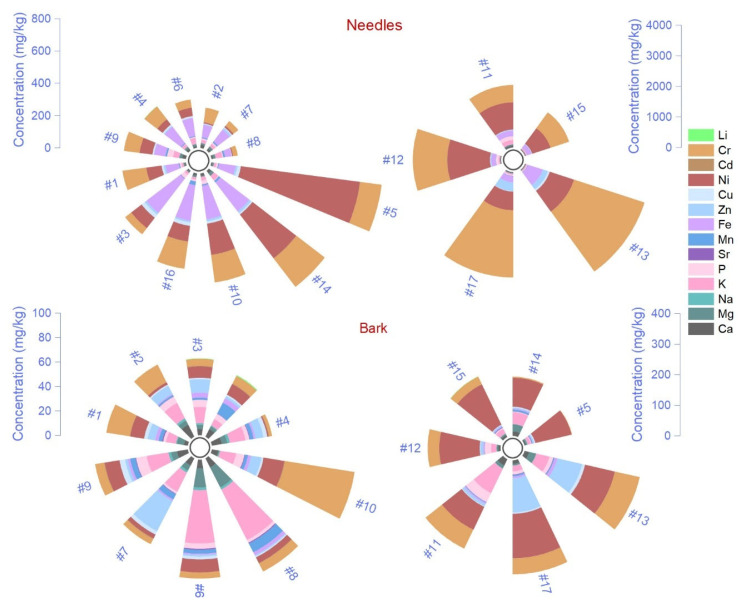
Bio-concentration factor (BCF) index illustration for each metal and altitude in bark and Norway spruce needles. See sites’ codes and altitudes in Figure 1.

**Figure 6 plants-14-00968-f006:**
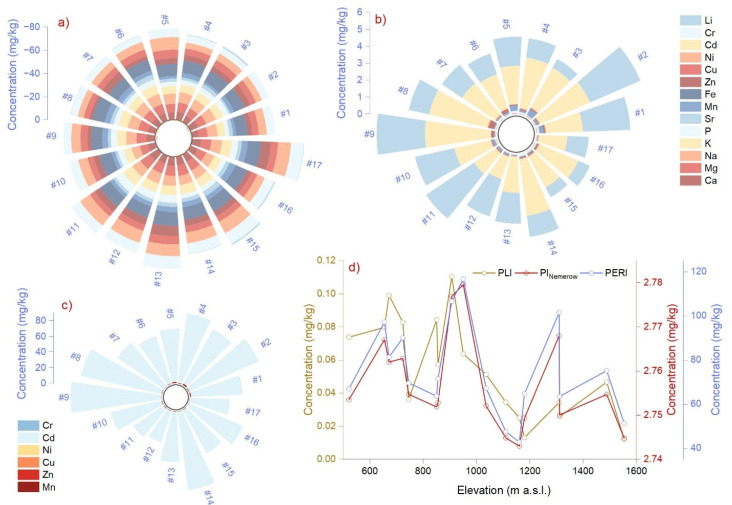
Graphical representation of environmental health risk assessment indices for each metal and altitude; (**a**) Geo-accumulation index (I_geo_), (**b**) Contamination factor (C*_f_*), (**c**) Ecological risk (E*_r_*), (**d**) Pollution load index (PLI), Nemerow pollution index (PI_Nemerow_), and potential ecological risk (PERI). See sites’ codes and altitudes in Figure 1.

**Figure 7 plants-14-00968-f007:**
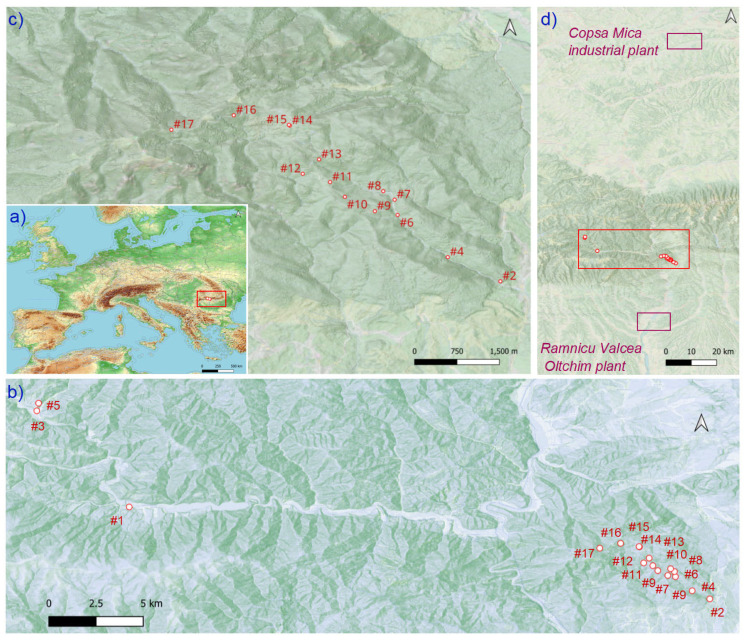
Location map of the studied area in (**a**) eastern Europe, (**b**) locations of the sampling sites in the Cozia Mountains, (**c**) a detailed illustration of the altitudinal gradient, and (**d**) distance and relief between study sites (red rectangle) and two industrial contamination sources (pink rectangles). The numbers on the map indicate altitude as follows: control sites [#1 (521 m), #3 (671 m), #5 (745 m)], and the altitudinal gradient used to evaluate the pollution status [#2 (654 m), #4 (723 m), #6 (850 m), #7 (855 m), #8 (907 m), #9 (950 m), #10 (1036 m), #11 (1110 m), #12 (1160 m), #13 (1180 m), #14 (1310 m), #15 (1311), #16 (1488 m), and #17 (1554 m)].

**Table 1 plants-14-00968-t001:** Summary statistics (mean ± standard error of the mean) for each metal investigated in Norway spruce needles, bark, and litter (data are given in mg kg^−1^), respectively. Statistical differences between needles, bark, and litter are indicated by different letters.

Element	Tissue or Component			Bonferroni Test			Levene’s Test	*RMSE*
	Needles	Bark	Litter	Bark vs. Needles *t*-Value	Litter vs. Needles *t*-Value	Litter vs. Bark *t*-Value	Prob > *F*	
K	6263 ± 133 (A)	4448 ± 339 (B)	250 ± 30 (C)	−6.06 ***	−20.11 ***	−14.04 ***	<0.0001	871
Ca	5391 ± 182 (A)	2692 ± 121 (B)	359 ± 36 (C)	−14.87 ***	−27.73 ***	−12.86 ***	<0.0001	528
Mg	840 ± 21 (A)	454 ± 71 (B)	70 ± 8.5 (C)	−6.29 ***	−12.55 ***	−6.25 ***	<0.0001	178
P	597 ± 76 (A)	199 ± 9.6 (B)	44 ± 6.5 (B)	−6.34 ***	−8.79 ***	−2.45	<0.0001	183
Na	81 ± 4.0 (B)	136 ± 8.3 (A)	120 ± 14 (A)	3.91 **	2.74 *	−1.17	0.08	41
Fe	4991 ± 376 (A)	94 ± 10 (B)	37 ± 4.4 (B)	−15.90 ***	−16.09 ***	−0.18	<0.0001	897
Mn	496 ± 48 (A)	219 ± 28 (B)	78 ± 14 (C)	−5.85 ***	−8.81 ***	−2.96 *	0.001	138
Zn	38 ± 4.86 (A)	23 ± 2.9 (B)	5.2 ± 1.37 (C)	−3.09 **	−6.98 ***	−3.88 ***	<0.0001	13
Cu	9.26 ± 1.1 (A)	1.78 ± 0.16 (B)	0.77 ± 0.07 (B)	−7.79 ***	−8.85 ***	−1.05	<0.0001	2.7
Sr	48.79 ± 4.3 (B)	39 ± 4.3 (B)	72 ± 8.5 (A)	−1.03	2.74 *	3.78 **	0.06	24
Cr	46 ± 6.4 (A)	4.05 ± 0.82 (B)	0.35 ± 0.07 (B)	−7.95 ***	−8.65 ***	−0.69	<0.0001	15
Ni	16 ± 2.29 (A)	2.22 ± 0.20 (B)	0.23 ± 0.06 (B)	−7.40 ***	−8.45 ***	−1.05	<0.0001	5.4
Li	16 ± 2.66 (B)	2.3 ± 0.75 (C)	35 ± 4.38 (A)	−3.24 **	4.58 ***	7.82 ***	<0.0001	12
As	0.61 ± 0.04 (A)	—	—	—	—	—	—	—
Cd	0.24 ± 0.02 (A)	0.09 ± 0.01 (B)	0.24 ± 0.01 (A)	−6.01 ***	−0.09	5.91 ***	0.06	0.07

* Significance levels are *p* < 0.05 *, *p* < 0.01 **, *p* < 0.001 ***.

**Table 2 plants-14-00968-t002:** Summary statistics of ratios calculated for litter and Norway spruce needles. ANOVAs were used to test differences between litter and needles.

Mole Ratio	Tissue or Component	Summary Statistics				One-Way ANOVA	
		Mean	SD	Min	Max	*F*-Value	Prob > *F*
Ca/Mg	litter	5.34	1.26	2.78	7.68	9.57	0.004
	needles	6.42	0.70	4.60	7.25		
Ca/Na	litter	3.58	1.90	0.68	6.61	128.55	<0.0001
	needles	70.22	24.15	54.87	152.89		
Ca/P	litter	10.82	6.40	2.45	28.67	0.80	0.37
	needles	13.28	9.33	5.02	31.67		
Ca/K	litter	1.58	0.58	0.89	3.03	25.83	<0.0001
	needles	0.85	0.07	0.71	0.96		
Na/Mg	litter	0.09	0.01	0.04	0.12	33.73	<0.0001
	needles	2.07	1.40	0.81	5.72		
Mg/P	litter	2.19	1.39	0.46	4.62	0.083	0.77
	needles	2.05	1.43	0.87	4.83		
K/Na	litter	2.42	1.37	0.31	4.55	189.96	<0.0001
	needles	81.40	23.58	60.44	157.68		
K/Mg	litter	3.60	1.06	1.80	5.77	181.89	<0.0001
	needles	7.48	0.53	6.45	8.21		
Li/Ca	litter	0.13	0.16	0.02	0.72	11.76	0.001
	needles	0.002	0.002	0.00	0.008		
Li/Na	litter	0.35	0.21	0.07	0.82	4.57	0.04
	needles	0.21	0.16	0.05	0.66		

**Table 3 plants-14-00968-t003:** Criteria for interpreting environmental and ecological health risk indices.

Indices	Interval of Values	Interpretation
BCF	≤1	plants only absorb metals
	>1	plants have the potential for accumulation
Igeo	≤1	uncontaminated
	0 ≤ Igeo < 1	uncontaminated to moderately contaminated
	1 ≤ Igeo < 2	moderately contaminated
	2 ≤ Igeo < 3	moderately to heavily contaminated
	3 ≤ Igeo < 4	heavily contaminated
	4 ≤ Igeo < 5	heavily to extremely contaminated
	Igeo ≥ 5	extremely contaminated
Cf	Cf < 1	low contamination
	1 < Cf < 3	moderate contamination
	3 < Cf < 6	considerable contamination
	Cf > 6	very high contamination
PLI	PLI < 1	uncontaminated
	1 ≤ PLI < 2	uncontaminated to moderately contaminated
	2 ≤ PLI < 3	moderately to strongly contaminated
	PLI ≥ 3	strongly contaminated
PINemerow	≤0.7	uncontaminated
	0.7–1	danger range
	1–2	low contamination
	2–3	moderate contamination
	≥3	severe contamination
Eri	Eri < 40	low
	40 ≤ Eri < 80	moderate
	80 ≤ Eri < 160	considerable
	160 ≤ Eri < 320	high
	320 ≥ Eri	very high
PERI	RI < 150	moderate
	150 ≤ RI < 300	considerable
	300 ≤ RI < 600	very high
	600 ≥ RI	

## Data Availability

Data are contained within the article and Appendix A.

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
