# Peer review of "Heavy Metal Health Risk Assessment in Picea abies L. Forests Along an Altitudinal Gradient in Southern Romania"

_plants, 2025, doi:10.3390/plants14060968_

Round 1
Reviewer 1 Report
Comments and Suggestions for Authors
Based on the hypothesis (i) whether there is significant contamination threatening human and environmental health and (ii) whether there are possible imbalanced nutrient mole ratios reflecting forest dieback in the region Authors evaluated nutrients (Ca, Mg, Na, K, P), non-toxic metals (Sr, Mn, Fe, Zn, Cu), and toxic metals (Cd, As, Li, Cr, Ni) in litter and P. abies needles and bark. They used two isotopic ratios (206/207Pb, 87/86Sr) to investigate the possible origins of metals in tree organs and litter, and the mole ratios Ca/P, Mg/P, Ca/Mg, K/Mg, Ca/Na, Ca/P, Mg/P, Ca/K, Na/Mg, K/Na, Li/Ca, and Li/Na were calculated and used as proxies for trees physiological alterations. This research has a large amount of work and can provide data supports and useful references for the forest ecosystems risk assessment and air deposition effects on trees. Moreover, I have some suggestions.
Table 1 lacks information on which values ​​apply to litter, bark and needles.
In Figure 4, the graphs for litter are probably incorrectly given by giving the letters a, b instead of a, d.
Author Response
Reviewer 1
Based on the hypothesis (i) whether there is significant contamination threatening human and environmental health and (ii) whether there are possible imbalanced nutrient mole ratios reflecting forest dieback in the region Authors evaluated nutrients (Ca, Mg, Na, K, P), non-toxic metals (Sr, Mn, Fe, Zn, Cu), and toxic metals (Cd, As, Li, Cr, Ni) in litter and P. abies needles and bark. They used two isotopic ratios (206/207Pb, 87/86Sr) to investigate the possible origins of metals in tree organs and litter, and the mole ratios Ca/P, Mg/P, Ca/Mg, K/Mg, Ca/Na, Ca/P, Mg/P, Ca/K, Na/Mg, K/Na, Li/Ca, and Li/Na were calculated and used as proxies for trees physiological alterations. This research has a large amount of work and can provide data supports and useful references for the forest ecosystems risk assessment and air deposition effects on trees. Moreover, I have some suggestions.
Response: We are grateful for your willingness to review the manuscript and for your valuable recommendations and suggestions. All suggestions were performed, and a new corrected version is available. Thank you!
Table 1 lacks information on which values ​​apply to litter, bark and needles.
Response: The information request was added. Thank you!
In Figure 4, the graphs for litter are probably incorrectly given by giving the letters a, b instead of a, d.
Response: Thank you! We corrected the figure explanation.
Reviewer 2 Report
Comments and Suggestions for Authors
Dear Authors,
Please find my comments in the attached file.
Regards

Author Response
We are grateful for your willingness to review the manuscript and for your valuable recommendations and suggestions. All suggestions were implemented in the new revised manuscript. Please find attached a version with track changes, which includes all corrections. Thank you!
Reviewer: What is the effect of growth to the trees of polution it is not clear
Author's response: This manuscript does not evaluate tree growth. Here, we assessed elemental concentration in three matrices (meddles, bark, and litter) resulting from air deposition. We used Health Risk assessment indices to evaluate the ecosystem health and the index of bioaccumulation factor of heavy metals in plants. The title was revised as follows: “Heavy metal health risk assessment in Picea abies forests along an altitudinal gradient in South Romania”. Thank you!
Reviewer: What about the research problem aims and tasks?
Author's response: The Abstract was rewritten to the maximum of 200 words allowed. Thank you!
Reviewer: This conclusion is too general and could be written under many simmilar articles. Please be more concrete and provide general conclusion from your work
Author's response: We rewrote the sentence for clarity: “The methodology and results presented herein may serve as a reference for future studies and provide a foundation for policymakers to develop management strategies to mitigate heavy metal pollution in natural forest ecosystems”.
Reviewer: Belongs to methods.
Author's response: The paragraph from lines 72-74 was transferred to the methods.
Reviewer: Why not to add ANOVA analysis?
Author's response: We added the ANOVA analysis results in the Table 1.
Reviewer: What is the effect of polution to the growth of the plants? dont you have some indicator plants to discuss?
Author's response: We used only the Norway spruce needles, bark, and litter in the analysis to evaluate the elemental distribution of the elements. The entire discussion chapter was rewritten. Please find the final version enclosed in the clean version of the revised manuscript. Thank you!
Reviewer: It is dissapointing that you evaluated the polutants but did not analyzed the effect of polution to growth of trees. Calculated indices provide some information but , it is still hard to imagine real situation
Author's response: We have collected cores. However, the LA-ICP-MS measurements will be presented separately in a new manuscript, as there is much information regarding dendrochemistry and climate.
Reviewer: From how many trees samples were collected? Did all trees participated in the analysis or did you use the selection method? How these locasions were selected? You have to visualize the locations on the map I think it is easy to do. Did you use some of these locations for control? If some of them are close to the roads or some of them are far from polution it has to be indicated.
Author's response: All the information requested was added to the manuscript. Please find the final version enclosed in the clean version of the revised manuscript. Thank you!
Reviewer: Looking to your research sampling plots it is obvious that they are concentrated in one area for some reason, why? only three plots are located little furwer. Why dont you have the research gradient, meaning spatial versatile distribution of the plots, closer and further from polution areas? Which are your control plots, to compare the effect of polution source?
Author's response: We have added the requested information in the text and changed the figure. The altitudinal gradient was firstly considered on the same slope; sincerely, we did not think about the versatile distribution of the plots. We initially selected three control plots in the forest on map #1, #3, and #5 and the altitudinal gradient as polluted. The results indicate no significant differences between unpolluted and altitudinal gradient.
Reviewer: Why six indices? what are the differences between them, dont the presented information overlap?
Author's response: We used a suite of indices because each had a different calculation base. Some differences were observed between indices applied to the litter, such as the contamination factor versus Igeo, Cf, PLI, PINemerow, and PERI. The bioconcentration factor also offers indices regarding a plant's ability to assimilate different metals in bark and needles.
Reviewer: What is extreemly high? Compared with what? This is not the conclusion but discussion unless you present the control value.
Author's response: We detailed the recommended limits in the discussion chapter and rewrote the Conclusion chapter. Thank you!

Reviewer 3 Report
Comments and Suggestions for Authors
General recommendations and questions
Abstract
Line 21-23. “The extremely high mole ratio, such as K/Na and Ca/Na mole, also documented anthropogenic origins and even illustrated possible dysfunctions in plant physiological processes.” How could low Na content negatively affect plant physiological processes? Na is not an essential nutrient and it is only welcome that K and Ca content is high relative to Na.
Line 26-28. “The health risk assessment classifies litter from uncontaminated to considerably contaminated at various altitudes, with a high ecological risk induced by Cd pollution mainly at 907, 950, and 1,310 m”. How high was the Cd contamination (concentration range), did the concentrations exceed the permissible limits?
Line 30-31. “The presented methodology and results provide a fundamental basis to mitigate heavy metal pollution in forest ecosystems.” Explain in more detail how this could reduce pollution in forests.
Introduction
In general, the introduction clearly reflects the research topic and the purpose of the work.
Materials and Methods
This is one of the most important sections in any scientific article, so it requires special accuracy and clarity. But on the other hand, overly detailed coverage of standard procedures also unnecessarily lengthens the article. References would be enough there. No need to go into detail about sample digestion procedures, ICP and AAS methodologies for element determination. A clear example of how not to describe every step taken: “Before each series of measurements, a calibration line was created for each element.”
At the same time, information about sample collection and preparation is poorly described and several questions arise: where was the tree bark collected - from the trunk or branches? How were the litter samples taken?
There is Figure 1 in the text, and Figure 8 for the figure itself. It is not correctly shown which areas were uncontaminated and which were contaminated. We can only guess.
4.2.1. Elemental analysis – “Samples were stored into plastic bags, air-dried, and mixed to achieve homogeneity and sieved (< 0.2 mm)” - How was it technically possible to sift through such a fine sieve if the samples were not ground?
Although two companies are listed as potential sources of pollution, no information is provided about what these factories are, what products they produce or process? What type of pollution would be expected?
4.4. Risk assessment
“The intervals of interpretation are presented separately for each index in the supplementary material (Table 1SM).” - Since I don't have supplementary materials at my disposal, I can't judge anything’ about them. Was it compared to any permissible limits, ranges?
Results
2.1. Heavy metals distribution and altitudinal variability in bark, needles and litter
This chapter provides information not only about heavy metals, but also about nutrients. This also applies to the title of Figure 2.
The data in Table 1 does not match the data in Figure 2. It is also not clear what data are presented in the 3 columns of the table: needles, bark, litter, or is there another order. For example, the highest P concentrations were in needles (Figure 2), but in Table 1, the highest concentrations are given in the second column – bark??? You can't really understand anything if the table doesn't indicate what is what in headings. This also applies to other elements.
Were Fe concentrations as high as 4,000, 6,000 mg/kg really found in needles?
2.2. Isotopic signature and mole ratios of Ca, K, Na, Mg, P, Li
Line 126-127. “The 87/86 Sr also decreased from needles > bark > litter (0.75 ± 0.34; 0.73 ± 0.02; 0.71 ± 0.00”. Can it really be assumed that 87/86Sr decreased from needles > bark > litter, if the values ​​are within the error limits? All these values are not different!
“The elements Ca and P had high concentrations in needles and were relatively lower in the litter.” How correct is it to talk about Ca and P pollution, because they are nutrients and the concentrations in spruce needles were not excessively high? On the contrary, the determined Ca results in needles correspond to generally accepted sufficiency ranges for spruce, but for P – there is a deficiency.
2.4. Environmental risk assessment
In general, when describing and evaluating the results, the authors should keep in mind that needles, as a vital, living, photosynthetic part of the tree, will contain higher concentrations of most elements than bark, which no longer participates in life functions, and of course higher than litter, which is actually the dead, inanimate part. Before litter is mineralized, the plant can absorb little.
Discussion
The Discussion is the most unconvincing part of this article.
Line 205-206. “The presented results show a disproportioned ratio between mineral concentration in needles vs. bark vs. litter.” How do you know that the ratio is disproportionate? Is there a rule somewhere that says what it should be?
You write that this indicates that HM pollution comes from air deposition, but it is most likely normal that there is more K in the needles than in the bark.
Line 234-235. What are you talking about in this sentence - your own research or that of others, if there is already a reference.
Line 236-237. An incomprehensible sentence and in general a lot of unnecessary information in this and next paragraph that is not related to the research.
3.2. Relationship between nutrients mole ratio and toxic metals
Line 259. Are these your found relationships or reference relationships?
Is a high ratio of Ca and K to Na a bad or good phenomenon? From the text of the article it is clear that you considered it undesirable. However, Na is undesirable for plants, but K and Ca are essential nutrients!!! Moreover, looking at the results, it is clear that the Ca and K content in the needles was not elevated. Similarly - would it be a problem if there was an extremely low Na/Mg (0.09)? That's fine, Na toxicities is not expected.
Line 266-267. “The K/Mg (7.4), Ca/P (13.2), and Ca/Mg (6.4) are also overestimated even for agricultural soils amended with supplements [63].” What does this have to do with soil? We're talking about needles, right?
Line. 268-270. “On the contrary, we found extremely low Na/Mg (0.09), Ca/K (0.85), and Mg/P (2.05) ratios in needles. Both concentrations and imbalanced ratios can be detrimental to plants since they affect biochemical processes and induce plasma-membrane inquiry, resulting in reduced growth mass deposition [64,65].”- Completely illogical and contradictory, high Na is not necessary and why is there a reference? Have these ratios been determined by other researchers? It is necessary to clearly distinguish the results of this study from other research results.
3.3. Environmental health risks
Line 278-282. What does this have to do with this study - clouds, temperature...? There is no need to fill a certain number of pages with text, the main thing to discuss is - was there pollution stated or not, did the remote areas (control) significantly differ from the park areas. Were the plants sufficiently vital and well supplied with nutrients, did the concentrations of heavy metals exceed the permissible norms, is this area a safe recreation area for people?
“…low transfer from litter in plant organs…” - Are you sure that it is possible - for elements from litter to efficiently and intensively transfer to plant parts? However, a rather time-consuming intermediate stage is required here - decomposition and release of elements for their uptake by the plant.
Conclusion
The conclusions should give answers to the purpose of the research and the questions rose in the tasks.
The aim of the research: “We chose this location to assess (i) whether there is significant contamination threatening human and environmental health and (ii) whether there are possible imbalanced nutrient mole ratios reflecting forest dieback in the region. To fulfill these aims, we evaluated nutrients (Ca, Mg, Na, K, P), non-toxic metals (Sr, Mn, Fe, Zn, Cu), and toxic metals (Cd, As, Li, Cr, Ni) in litter and P. abies needles and bark. Two isotopic ratios (206/207Pb, 87/86Sr) were used to investigate the possible origins of metals in tree organs and litter. The mole ratios Ca/P, Mg/P, Ca/Mg, K/Mg, Ca/Na, Ca/P, Mg/P, Ca/K, Na/Mg, K/Na, Li/Ca, and Li/Na were calculated and used as proxies for trees physiological alterations. Finally, we performed an ecosystem health risk assessment to document air deposition effects on trees and deliver information on ecological integrity and ecosystem service demand in the study area.”
Also, conclusions should provide information, a conclusion, hypothesis on how the knowledge gained in this study could be useful in the future, in practice.
I recommend to rewrite the conclusions, giving a clear and research-based answer to the raised questions. I don't find it in the existing conclusions. I'm not sure that based on the information and interpretation in the article, it will be easy to provide such. Many things should be reviewed and the emphasis changed.
Did you really find significant pollution that poses a threat to human and environmental health? How did you assess it? What was your basis for this? Were there any significant problems with the forests in the region, or were the trees severely damaged? The reader has no information about this. Was there any correlation between the element content in the needles and the state of tree vitality?
Concluding remarks.
Overall, the article leaves a dual impression. On the one hand, the topic of environmental pollution has been relevant both globally and locally for a long time, the article contains new knowledge and quite extensive data material. On the other hand, it seems that everything is mixed together - elements necessary for vital tree growth and polluting elements. The assessment of their concentrations is mechanical - more, less, higher, lower; without understanding plant physiology, mineral nutrition, optimal and harmful levels. No comparisons were made with nutrient sufficiency levels for spruce needles, permissible levels for HM, or whether there was any difference between the study sites and the control (uncontaminated) sites.
Therefore, it is necessary to make significant corrections. Relatively broad suggestions and recommendations for improving the quality of the article are provided in the comments to the authors.
Author Response
General recommendations and questions
We agree with all remarks, corrections, and recommendations, for which we express our sincere gratitude. We also thank you for accepting our manuscript for review. All manuscript chapters were partially and totally (Discussion) rewritten according to the reviewer's instructions. All changes are emphasized with track changes in a separate file.
Abstract
Line 21-23. “The extremely high mole ratio, such as K/Na and Ca/Na mole, also documented anthropogenic origins and even illustrated possible dysfunctions in plant physiological processes.”
Reviewer: How could low Na content negatively affect plant physiological processes?
Author's response: The sentence was revised, and the Abstract was rewritten. Thank you!
Reviewer: Na is not an essential nutrient and it is only welcome that K and Ca content is high relative to Na.
Author's response: The following paper carefully detailed this paradigm: “Brown, P. H., Zhao, F. J., & Dobermann, A. (2022). What is a plant nutrient? Changing definitions to advance science and innovation in plant nutrition. Plant and Soil, 476(1), 11-23.”. According to the “Marschner P (ed) (2012) Marschner’s mineral nutrition of higher plants, 3rd edn. Academic, Amsterdam”, EU regulation mentions Na as an essential element in relatively small amounts for plant growth. Other studies separated Na from “essential nutrients” in a special case of “functional elements” “Subbarao, G. V., Ito, O., Berry, W. L., & Wheeler, R. M. (2003). Sodium—a functional plant nutrient. Critical Reviews in Plant Sciences, 22(5), 391-416”. In the Introduction chapter, we discussed the above literature. Thank you!
Line 26-28. “The health risk assessment classifies litter from uncontaminated to considerably contaminated at various altitudes, with a high ecological risk induced by Cd pollution mainly at 907, 950, and 1,310 m”.
Reviewer: How high was the Cd contamination (concentration range), did the concentrations exceed the permissible limits?
Author's response: We added detailed information regarding the Cd concentration range. The safety limits of 0.098 mg/kg indicated by the Heinrichs et al 1980 in lithosphere were exceeded (Heinrichs, H., Schulz-Dobrick, B., & Wedepohl, K. H. (1980). Terrestrial geochemistry of Cd, Bi, Tl, Pb, Zn and Rb. Geochimica et Cosmochimica Acta, 44(10), 1519-1533).
Line 30-31. “The presented methodology and results provide a fundamental basis to mitigate heavy metal pollution in forest ecosystems.”
Reviewer: Explain in more detail how this could reduce pollution in forests.
Author's response: We rewrote the sentence for clarity: “The methodology and results presented herein may serve as a reference for future studies and provide a foundation for policymakers to develop management strategies to mitigate heavy metal pollution in natural forest ecosystems”.
Introduction
Reviewer: In general, the introduction clearly reflects the research topic and the purpose of the work.
Author's response: Thank you! We added several statements regarding the definition of essential minerals, functional minerals, and toxic metals, and we reorganized the introduction for more clarity.
Materials and Methods
This is one of the most important sections in any scientific article, so it requires special accuracy and clarity. But on the other hand, overly detailed coverage of standard procedures also unnecessarily lengthens the article. References would be enough there. No need to go into detail about sample digestion procedures, ICP and AAS methodologies for element determination. A clear example of how not to describe every step taken: “Before each series of measurements, a calibration line was created for each element.”
Reviewer: At the same time, information about sample collection and preparation is poorly described and several questions arise: where was the tree bark collected - from the trunk or branches? How were the litter samples taken?
Author's response: We improved the material and method section and added new information regarding sample collection to the manuscript as follows: “The sampling design was created to understand whether pollution differs at an altitudinal gradient. In mid-October 2021, we collected first and second-year-old needles and tree bark from dominant, apparently healthy P. abies trees. In each sampling plot, samples were collected from three trees. Specifically, three plots were selected from the litter, with a distance of five meters between each plot. All materials were subsequently mixed, and a selection was further evaluated. Of the 17 locations, 3 are outside the national park inside the mountain with no evident contamination factor considered as control sites and 14 within the park near a national highly used car road (Figure 1c).”
Reviewer: There is Figure 1 in the text, and Figure 8 for the figure itself. It is not correctly shown which areas were uncontaminated and which were contaminated. We can only guess.
Author's response: All figure numbers were revised. Figure 1 was revised, and the new figure shows sites in the altitudinal transect.
4.2.1. Elemental analysis – “Samples were stored into plastic bags, air-dried, and mixed to achieve homogeneity and sieved (< 0.2 mm)”
Reviewer: How was it technically possible to sift through such a fine sieve if the samples were not ground?
Author's response: We corrected the statement as follows: “The collected samples were transported into plastic bags and air-dried at the laboratory at 80°C for 24 h, and the dry weight was measured. The material was finely ground in an agate mortar before being sieved (< 0.2 mm).”
Reviewer: Although two companies are listed as potential sources of pollution, no information is provided about what these factories are, what products they produce or process? What type of pollution would be expected?
Author's response: The following paragraph was added: “Oltchim chemical plant was founded in 1966 to produce inorganic, macromolecular, and organic synthesis products (chlor-alkali, oxo-alcohols, propylene oxide, vinyl chloride monomer). This industrial complex includes the Sodic Govora plants that produced live steam, which contain facilities for producing, conditioning, and delivering fuels, water, electricity, and thermal energy. The principal emissions of metal pollution resulting from its activities include Cu, Zn, Pb, Mn, Cd, Ni, Co, Fe, and Hg. The industrial activity in Copsa Mica began in 1935 and has caused significant environmental issues by releasing substantial amounts of Hg, Sr, Li, Pb, Cd, Cu, Al, Cr, Ni, Co, Fe, As, Zn, and Mn into the atmosphere, earning recognition as the most polluting source in Europe”.
4.4. Risk assessment
“The intervals of interpretation are presented separately for each index in the supplementary material (Table 1SM).”
Reviewer: Since I don't have supplementary materials at my disposal, I can't judge anything’ about them. Was it compared to any permissible limits, ranges?
Author's response: To clearly understand the results, the table outlining the permissible limits for each index was included within the main body of the manuscript. Thank you!
Results
2.1. Heavy metals distribution and altitudinal variability in bark, needles and litter
Reviewer: This chapter provides information not only about heavy metals, but also about nutrients. This also applies to the title of Figure 2.
Author's response: For clarity, several general aspects of the manuscript were improved:
- The title of the manuscript was changed “Heavy metal health risk assessment in Picea abies forests along an altitudinal gradient in South Romania”
- In the Introduction, Results, and Discussion, we separated the metal pollution from essential elements.
- After rearranging the figure numbers, Figure 2 was Figure 1, and the caption was corrected as follows: “Elemental chemical concentration in Norway spruce needles, bark, and litter measured in Cozia”.
Reviewer: The data in Table 1 does not match the data in Figure 2. It is also not clear what data are presented in the 3 columns of the table: needles, bark, litter, or is there another order. For example, the highest P concentrations were in needles (Figure 2), but in Table 1, the highest concentrations are given in the second column – bark??? You can't really understand anything if the table doesn't indicate what is what in headings. This also applies to other elements.
Author's response: Table 1 was improved according to recommendations. Thank you!
Reviewer: Were Fe concentrations as high as 4,000, 6,000 mg/kg really found in needles?
Author's response: Yes, we measured Fe range concentrations (mg/kg) of 1000-6599 (needles), 45-173 (bark), and 12-69 (litter).
2.2. Isotopic signature and mole ratios of Ca, K, Na, Mg, P, Li
Reviewer: Line 126-127. “The 87/86 Sr also decreased from needles > bark > litter (0.75 ± 0.34; 0.73 ± 0.02; 0.71 ± 0.00”. Can it really be assumed that 87/86Sr decreased from needles > bark > litter, if the values ​​are within the error limits? All these values are not different!
Author's response: We rephrased as following: “The 206/207Pb values had a range interval of 1.16-1.20 (needles), 1.15-1.17 (bark), and 1.24-1.16 (litter), respectively, with mean values of 1.18 ± 0.01; 1.16 ± 0.00; 1.14 ± 0.01 (Figure 2). The 87/86Sr values varied between 0.71-0.83 (needles), 0.71-0.78 (bark), and 0.71-0.73 (litter), and the average was 0.75 ± 0.34; 0.73 ± 0.02; 0.71 ± 0.00”.
Reviewer: “The elements Ca and P had high concentrations in needles and were relatively lower in the litter.” How correct is it to talk about Ca and P pollution, because they are nutrients and the concentrations in spruce needles were not excessively high? On the contrary, the determined Ca results in needles correspond to generally accepted sufficiency ranges for spruce, but for P – there is a deficiency.
Author's response: The sentence was deleted, and a paragraph was added in the discussion chapter, which evaluated the range of all minerals in relation to recommended limits. Please see the 3.1. Elemental distribution, origins and altitudinal trends subchapter. Thank you!
2.4. Environmental risk assessment
Reviewer: In general, when describing and evaluating the results, the authors should keep in mind that needles, as a vital, living, photosynthetic part of the tree, will contain higher concentrations of most elements than bark, which no longer participates in life functions, and of course higher than litter, which is actually the dead, inanimate part. Before litter is mineralized, the plant can absorb little.
Author's response: Thank you for the recommendation. We partially rewrote the subchapter 2.4 Environmental risk assessment to add more clarity.
Discussion
Reviewer: The Discussion is the most unconvincing part of this article.
Author's response: The discussion chapter was rewritten. Thank you for your suggestions and recommendations, which helped us improve our work.
Reviewer: Line 205-206. “The presented results show a disproportioned ratio between mineral concentration in needles vs. bark vs. litter.” How do you know that the ratio is disproportionate? Is there a rule somewhere that says what it should be? You write that this indicates that HM pollution comes from air deposition, but it is most likely normal that there is more K in the needles than in the bark.
Author's response: The whole paragraph was rewritten. Thank you!
Reviewer: Line 234-235. What are you talking about in this sentence - your own research or that of others, if there is already a reference.
Author's response: We have rephrased as flowing “The presented results indicated high bioavailability in Norway spruce, mainly for Zn, which decreases with distance from a point source. These findings were also presented in other similar studies [45]”.
Reviewer: Line 236-237. An incomprehensible sentence and in general a lot of unnecessary information in this and next paragraph that is not related to the research.
Author's response: The following sentences were deleted. Thank you!
3.2. Relationship between nutrients mole ratio and toxic metals
Reviewer: Line 259. Are these your found relationships or reference relationships?
Author's response: We have delimited our results from other studies. Thank you. The sentence was rewritten as follows: “Unfortunately, few studies on tree species in natural forests evaluate the toxicity of the deficiency of various mole ratios compared with stoichiometry in Norway spruce, with more results reported for Larix sp. and Pinus sp [25,99,100]. Even so, literature indicates an optimal mole ratio of Ca/Mg (6.5:1), Ca/K (13:1), Mg/K (2:1), Ca/P (1-2:1), K/Mg (1.5:1) [101-103]”.
Reviewer: Is a high ratio of Ca and K to Na a bad or good phenomenon? From the text of the article it is clear that you considered it undesirable. However, Na is undesirable for plants, but K and Ca are essential nutrients!!! Moreover, looking at the results, it is clear that the Ca and K content in the needles was not elevated. Similarly - would it be a problem if there was an extremely low Na/Mg (0.09)? That's fine, Na toxicities is not expected.
Author's response: The whole subchapter was rewritten; please find the last version enclosed in the revised manuscript.
Reviewer: Line 266-267. “The K/Mg (7.4), Ca/P (13.2), and Ca/Mg (6.4) are also overestimated even for agricultural soils amended with supplements [63].” What does this have to do with soil? We're talking about needles, right?
Author's response: The sentence was deleted. Thank you!
Reviewer: Line. 268-270. “On the contrary, we found extremely low Na/Mg (0.09), Ca/K (0.85), and Mg/P (2.05) ratios in needles. Both concentrations and imbalanced ratios can be detrimental to plants since they affect biochemical processes and induce plasma-membrane inquiry, resulting in reduced growth mass deposition [64,65].”- Completely illogical and contradictory, high Na is not necessary and why is there a reference? Have these ratios been determined by other researchers? It is necessary to clearly distinguish the results of this study from other research results.
Author's response: The sentence was rewritten. Thank you!
3.3. Environmental health risks
Reviewer: Line 278-282. What does this have to do with this study - clouds, temperature...? There is no need to fill a certain number of pages with text, the main thing to discuss is - was there pollution stated or not, did the remote areas (control) significantly differ from the park areas. Were the plants sufficiently vital and well supplied with nutrients, did the concentrations of heavy metals exceed the permissible norms, is this area a safe recreation area for people?
Author's response: The Discussion chapter has been revised to maintain a cohesive narrative emphasizing the concentration of metals in relation to other studies and established thresholds. Additionally, we have distinguished the control sites from those located in the altitudinal transect, which are suspected of being influenced by pollution, and indicated that no significant differences were detected between these groups. The evaluation of toxic metals was conducted to explore the probable anthropogenic sources of contamination arising from two industrial zones and the roadway that traverses the area pack.
Reviewer: “…low transfer from litter in plant organs…” - Are you sure that it is possible - for elements from litter to efficiently and intensively transfer to plant parts? However, a rather time-consuming intermediate stage is required here - decomposition and release of elements for their uptake by the plant.
Author's response: The sentence was revised. Thank you!
Conclusion
The conclusions should give answers to the purpose of the research and the questions rose in the tasks.
Reviewer: The aim of the research: “We chose this location to assess (i) whether there is significant contamination threatening human and environmental health and (ii) whether there are possible imbalanced nutrient mole ratios reflecting forest dieback in the region. To fulfill these aims, we evaluated nutrients (Ca, Mg, Na, K, P), non-toxic metals (Sr, Mn, Fe, Zn, Cu), and toxic metals (Cd, As, Li, Cr, Ni) in litter and P. abies needles and bark. Two isotopic ratios (206/207Pb, 87/86Sr) were used to investigate the possible origins of metals in tree organs and litter. The mole ratios Ca/P, Mg/P, Ca/Mg, K/Mg, Ca/Na, Ca/P, Mg/P, Ca/K, Na/Mg, K/Na, Li/Ca, and Li/Na were calculated and used as proxies for trees physiological alterations. Finally, we performed an ecosystem health risk assessment to document air deposition effects on trees and deliver information on ecological integrity and ecosystem service demand in the study area.”
Author's response: We rewrote the conclusion chapter.
Reviewer: Also, conclusions should provide information, a conclusion, hypothesis on how the knowledge gained in this study could be useful in the future, in practice.
Author's response: We stated, according to our findings, that ”Norway spruce can bioaccumulate high concentrations of Fe, Zn, Cu, Ni, and Cr in needles and bark, which is why it can be used for bioremediation strategies. ”
Reviewer: I recommend to rewrite the conclusions, giving a clear and research-based answer to the raised questions. I don't find it in the existing conclusions. I'm not sure that based on the information and interpretation in the article, it will be easy to provide such. Many things should be reviewed and the emphasis changed.
Author's response: We rewrote the conclusion chapter.
Reviewer: Did you really find significant pollution that poses a threat to human and environmental health? How did you assess it? What was your basis for this? Were there any significant problems with the forests in the region, or were the trees severely damaged? The reader has no information about this. Was there any correlation between the element content in the needles and the state of tree vitality?
Author's response: First, we eliminated human health references since no such indices were estimated. Regarding environmental health, we evaluated health risk assessment indices, which can offer clues regarding environmental status health. From this area, we collected tree cores that will be assessed separately in a study focusing on dendrochemistry.
Concluding remarks.
Reviewer: Overall, the article leaves a dual impression. On the one hand, the topic of environmental pollution has been relevant both globally and locally for a long time, the article contains new knowledge and quite extensive data material. On the other hand, it seems that everything is mixed together - elements necessary for vital tree growth and polluting elements. The assessment of their concentrations is mechanical - more, less, higher, lower; without understanding plant physiology, mineral nutrition, optimal and harmful levels. No comparisons were made with nutrient sufficiency levels for spruce needles, permissible levels for HM, or whether there was any difference between the study sites and the control (uncontaminated) sites. Therefore, it is necessary to make significant corrections. Relatively broad suggestions and recommendations for improving the quality of the article are provided in the comments to the authors.
Author's response: We agreed with all remarks, corrections, and recommendations, for which we express our sincere gratitude. Thank you!

Round 2
Reviewer 2 Report
Comments and Suggestions for Authors
Dear Authors,
I have reviewed the presented second version of the manuscript. The paper is well written, well structured with presented nice graphs and useful information. However, from my point of view, the paper moved very much to chemical part. Thus, what I do lack in this paper is analysis of response of plants to the pollution. Because of this we can not judge the possible effect of pollution to the plants. May be plants can tolerate well this amount of pollution or may be it is already deadly levels. Second issue authors clamed that they evaluate only the effect of altitude. Thats ok, but to analyze the effect of altitude without taking into account the variation of distance from pollution source, well it is like to hide half of the picture and show only another half of picture. I hope that these my concerns will be evaluated by academic editor. >From my side I conclude that the paper could be published, without any changes.
Regards
Author Response
Dear Authors,
I have reviewed the presented second version of the manuscript. The paper is well written, well structured with presented nice graphs and useful information. However, from my point of view, the paper moved very much to chemical part. Thus, what I do lack in this paper is analysis of response of plants to the pollution. Because of this we can not judge the possible effect of pollution to the plants. May be plants can tolerate well this amount of pollution or may be it is already deadly levels. Second issue authors clamed that they evaluate only the effect of altitude. Thats ok, but to analyze the effect of altitude without taking into account the variation of distance from pollution source, well it is like to hide half of the picture and show only another half of picture. I hope that these my concerns will be evaluated by academic editor. >From my side I conclude that the paper could be published, without any changes.
Dear reviewer,
Thank you for your time, involvement, and dedication in evaluating our manuscript. We learned the suggestion, and the next sampling experiment will adopt this strategy.
Sincerely yours,
Corresponding author
Reviewer 3 Report
Comments and Suggestions for Authors
The authors have done a lot of work and undeniably improved the quality of the article. However, some minor aspects still need improvement.
Line 98-99. “The average elemental chemical concentration showed no significant differences between control sites and the altitudinal transect assessed.” If this is the case, then either the control sites were not chosen correctly, or the conclusions about anthropogenic pollution associated with the factories mentioned in the study are not really understandable/correct. This should be explained and analyzed.
Line 106-108. “The heavy metal analysis of the bark indicates lower concentrations than in the needles, which can be used to conclude that needles can be used as bioindicators of heavy metal pollution and the status of elements essential for vital growth [30,31].” The Results section does not need to discuss the results obtained, that is a Discussion task. It is completely incomprehensible why there is a reference at the end of the sentence. Is that your conclusion, or someone else's?
Line 479-480. “To evaluate HM content in the litter, we collected leaves and branches (< 5 cm diameter) from the northern part of the tree trunk.” This is completely incomprehensible. And Picea abies don't have leaves!
Author Response
The authors have done a lot of work and undeniably improved the quality of the article. However, some minor aspects still need improvement.
Author's Response: We express our sincere gratitude and appreciate the time spent on the evaluation. All remarks were of great value and significantly improved our work. Thank you!
Line 98-99. “The average elemental chemical concentration showed no significant differences between control sites and the altitudinal transect assessed.” If this is the case, then either the control sites were not chosen correctly, or the conclusions about anthropogenic pollution associated with the factories mentioned in the study are not really understandable/correct. This should be explained and analyzed.
Author's Response: We will take the third option: the pollution transported by air mases cannot be separated at small scales. Yes, your point of view is correct, and in the next step, we will choose a new experimental design consisting of sampling plots located in a grid with longer distances to cover a larger area. There is pollution since a new manuscript, which will be under review in the following days, was focused in the vicinity of the Copsa Mica Industrial area, and the results can be compared with this study. We added a correction in the manuscript as folowing: "The average elemental chemical concentration showed no significant differences between control sites and the altitudinal transect assessed, documenting that the pollution resulted from atmospheric pollution cover a wide spatial range".
Line 106-108. “The heavy metal analysis of the bark indicates lower concentrations than in the needles, which can be used to conclude that needles can be used as bioindicators of heavy metal pollution and the status of elements essential for vital growth [30,31].” The Results section does not need to discuss the results obtained, that is a Discussion task. It is completely incomprehensible why there is a reference at the end of the sentence. Is that your conclusion, or someone else's?
Author's Response: Thank you. During the first stage of the revision process, we wanted to add Results and Discussion to the same chapter. In the end, we considered your recommendation to separate our results from previous results found in the literature, and we split the text again into distinct chapters. The sentence was rephrased for clarity and was added to the discussion chapter. The sentence was paraphrased as follows: "Our results show that the heavy metal analysis of the bark had lower concentrations than in the needles, which can be used to conclude that needles can be used as bioindicators of heavy metal pollution and the status of elements essential for vital growth. Similar results were previously presented in literature [30,31]. "
Line 479-480. “To evaluate HM content in the litter, we collected leaves and branches (< 5 cm diameter) from the northern part of the tree trunk.” This is completely incomprehensible. And Picea abies don't have leaves!
Author's Response: Thank you for carefully reading the manuscript and for your valuable help in improving it. We rephrased as follows: "To assess heavy metal (HM) content within the litter, we collected samples from the uppermost layer of the forest floor, which contains mineral organic debris, including needles, branches (less than 5 cm in diameter), fruits, and various types of plant matter exhibiting different stages of decomposition."
Thank you!,
Sincerely yours,
On behalf of all coauthors,
Corresponding author